# CONDITIONAL DIFFUSION INPAINTING FOR SKETCH-TO-FACE SYNTHESIS

Sanhita Pathak*[1], Vinay Kaushik[2], and Brejesh Lall[1]

[1]Indian Institute of Technology Delhi
[2]Indian Institute Of Information Technology Sonepat
Sanhita.Pathak@dbst.iitd.ac.in, vkaushik@iiitsonepat.ac.in, brejesh@ee.iitd.ac.in

## Abstract

Generating realistic faces from monochromatic sketches is challenging due to missing details like expressions and skin tones. GANs struggle with stability and structure, while diffusion models face issues with monochrome inputs and high costs. DSFace, a latent diffusion-based conditional inpainting framework, addresses these challenges using a frozen Paint-by-Example diffusion model with ControlNet conditioning and DINO-V2 embeddings from a GAN-generated coarse image. Trained on the CUFS dataset, DSFace achieves superior realism, perceptual quality, and structural alignment.

## 1 Introduction

Generating realistic human faces from monochrome sketches is a key challenge in image-to-image translation [7, 8], benefiting domains such as character design and forensic analysis. However, the narrow distribution of single-channel sketch data and the lack of semantic cues hinder robust feature extraction and generalization. Traditional GAN-based methods [9–11] leverage semantic masks or paired supervision but often lose fine details such as wrinkles, expressions, and accessories. Diffusion models (DM) [12] and CLIP [13] have shown promise in text-to-image synthesis, yet their direct adaptation to sketch-to-image translation remains limited due to weak structural correspondence. Conditioning-guided diffusion models like ILVR [14] and SDEdit [8] use RGB references for control but still fail to preserve accurate facial geometry.

We propose a latent diffusion-based framework trained on a sketch–face dataset that reformulates sketch-to-face synthesis as a conditional inpainting problem. By leveraging the conditioning flexibility of diffusion models, we employ coarse image guidance, A PBE based baseline and DINO-V2 embeddings to provide global structural consistency. Furthermore, ControlNet and parsed facial maps enable fine-grained control over local facial attributes. The proposed model demonstrates superior realism, structural alignment, and quantitative performance

*Corresponding Author.

on the CUFS dataset. Our contributions are as follows, 1) Reframe sketch-to-image generation as conditional inpainting using latent diffusion. 2) Employ DINO-V2 embeddings and GAN-generated coarse images for structural alignment. 3) Integrate ControlNet and semantic face parsing for fine-grained control.

## 2 Methodology

We propose solving sketch-to-face generation as a conditional diffusion inpainting task utilizing ControlNet encoder for structural control, and conditioning Dino-v2 embeddings from a coarse input generated by a GAN module.

The proposed methodology consists of two stages. In **Stage 1**, a Generative Adversarial Network (GAN) is employed to generate a coarse face image $I_{\text{coarse}}$ from the input facial sketch $I_s$. The resulting coarse image $I_{\text{coarse}}$ serves as a global structural prior, providing essential conditioning for the subsequent stage. The **Stage 2** consists of diffusion based pipeline, which consists of a frozen Diffusion Denoising Unet. The inputs to this module are face segmentation $I_{seg}$, face agnostic binary mask $I_{mask}$, face agnostic rgb image $I_{ag}$, input image with added noise $I_t$. The diffusion pipeline is guided by the structural conditioning from a ControlNet module for enabling the structural coherence with realistic facial features, the features guiding the controlNet are extracted from input sketch image $I_s$. An additional conditioning input that guides the ControlNet is the Dino-v2 embeddings computed from the provided coarse input image from stage 1, $I_{coarse}$. Dino-v2 module enables the pipeline to capture more detailed information about facial attributes, thereby enhancing control over the generation of facial regions. The ControlNet is trained to predict total noise, with the final clean generated face denoted as $I_{out}$.

## 3 Experiments

### 3.1 Implementation Details

We conduct end-to-end training using one NVIDIA A100 GPU with image resolution of $512 \times 384$. Training runs for 150 epochs using the AdamW optimizer

| Metrics | DualGAN [1] | Pix2Pix [2] | UGATIT [3] | NICE-GAN [4] | FRAN [5] | DiSS [6] | DCNP [7] | Ours |
|---|---|---|---|---|---|---|---|---|
| ↑SR-SIM | 0.8687 | 0.8681 | 0.8758 | 0.8800 | 0.8822 | 0.8655 | 0.8787 | **0.8941** |
| ↑MSSIM | 0.7938 | 0.7956 | 0.8090 | 0.8206 | 0.8381 | 0.8041 | 0.8181 | **0.8618** |
| ↓FID | 126.53 | 85.55 | 117.34 | 84.96 | 73.21 | 86.22 | 65.49 | **62.58** |
| ↑FSIM | 0.7893 | 0.7913 | 0.7955 | 0.8123 | 0.8187 | 0.7763 | 0.8121 | **0.8352** |
| ↑VIF | 0.1128 | 0.1098 | 0.1367 | 0.1427 | 0.1798 | 0.1253 | 0.1475 | **0.2321** |
| ↑IS | 1.32 | 1.35 | 1.42 | 1.41 | 1.35 | 1.34 | 1.40 | **1.49** |

**Table 1.** Comparison of various models using different metrics on CUSK datasets. Bold values indicate the best performance.

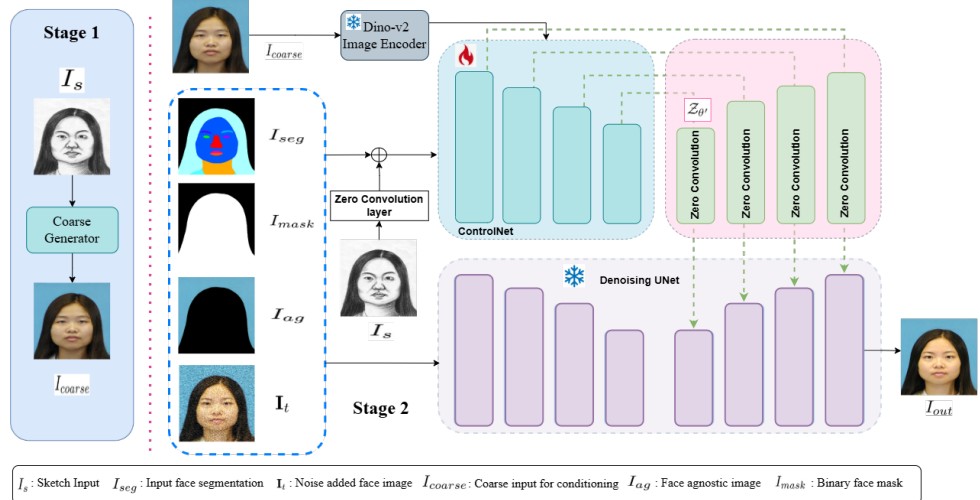

**Figure 1.** Architecture diagram of our proposed approach for sketch to face image synthesis

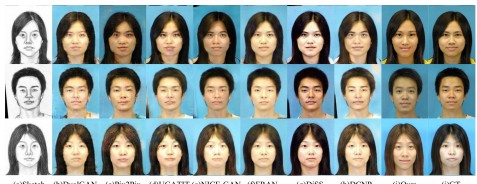

**Figure 2.** Qualitative comparison of our proposed approach with state-of-the-art approaches for sketch to face image synthesis.

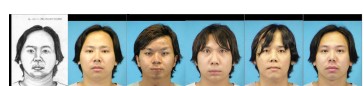

**Figure 3.** Qualitative Ablation of effect of various proposed modules on quality of generated faces.

with a learning rate of $2 \times 10^{-5}$. Experiments are performed on the CUFS dataset [15], which includes 606 identities from CUHK, AR, and XM2VTS, each containing one FPS pair, split in an 80:20 ratio for training and testing.

### 3.2 Results

We evaluate the proposed model for sketch-to-face synthesis through qualitative and quantitative comparisons with state-of-the-art methods, including DualGAN [1], Pix2Pix [2], UGATIT [3], NICE-GAN [4], FRAN [5], DiSS [6], and DCNP [7]. As shown in Figure 2, earlier methods often yield blurry results or artifacts, while our approach produces sharper, more realistic images with well-preserved facial features and textures, benefitting from DINO-v2 and ControlNet conditioning.

Quantitatively, Table 1 highlights our approach's superior performance, achieving the highest SR-SIM (0.8941), MSSIM (0.8618), FSIM (0.8352), VIF (0.2321), and IS (1.49), along with the lowest FID (62.58). These results confirm that our approach delivers enhanced realism, structural consistency, and perceptual fidelity compared to existing approaches.

### 3.3 Ablation Study

We evaluate our proposed method through quantitative and qualitative ablations (Figure 3). The baseline PBE model, conditioned only on sketch input, fails to preserve structural and perceptual fidelity, yielding poor FID, MSSSIM, and IS scores. Adding a ControlNet encoder enhances facial geometry consistency, while incorporating face segmentation improves perceptual quality through semantic guidance, though texture and color remain inconsistent. Finally, conditioning ControlNet with DINO-v2 embeddings from the coarse generated image provides refined structural and appearance details.

## 4 Conclusion

We propose a latent diffusion model that generates realistic faces from sketches using conditional inpainting with semantic guidance. It achieves state-of-the-art realism and fidelity on the CUFS dataset, advancing sketch-based synthesis applications.

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
