# OpenReview forum: "CONDITIONAL DIFFUSION INPAINTING FOR SKETCH-TO-FACE SYNTHESIS"
_NLDL.org/2026/Abstracts_Track — NLDL 2026 Abstracts_

### Official Review · Reviewer_1ALG · 2025-10-24

**Soundness:** 4
**Correctness:** 4
**Rating:** 4
**Confidence:** 2

**Summary:**

This work proposes conditioning a latent-diffusion model for synthesizing faces from sketches by leveraging DINO-V2 embeddings as semantic features for guiding the pre-trained generative model. The proposed approach considers first generating a coarse face image using a GAN and then guiding a diffusion pipeline with a ControlNet that takes such embeddings as input.

**Strengths:**

The idea of conditioning the diffusion based on semantic features is interesting and the example from the ablation study (Figure 3) clearly shows the benefit of this conditioning. The results also reveal that the proposed method is superior to other generative models on the CUSK dataset. The idea of using a coarse image as a structural prior in the two-stage modeling pipeline is interesting.

**Weaknesses:**

1. Only a single example (Figure 3) is shown, despite mentioning qualitative and quantitative ablation studies.
2. Please clarify what serves as the ControlNet input when Dino-V2 conditioning is removed.
3. The stage 1 is interesting, but given known training issues (e.g. instability) of GANs, it would help to explain why this approach was preferred over more stable generative models?

---

### Official Review · Reviewer_4Zx4 · 2025-10-24

**Soundness:** 3
**Correctness:** 3
**Rating:** 4
**Confidence:** 4

**Summary:**

The paper introduces a method for synthesizing realistically looking images from hand-drawn sketches.  The method levarages a GAN to generate a coarse-looking version of the image, which is further refined using the DINO-V2 embeddings of the image and a denoising U-Net for the diffusion process. This leads to increased performance across all metrics compared to the baselines.

**Strengths:**

The authors clearly state the methodology and motivation for the research. The method makes intuitive sense, is properly explained and appears valid.
The authors report that their proposed method outpeforms all baselines across all metrics, which in the field of generative image modeling is not a given.

The paper is well written and understandable.

**Weaknesses:**

The primary weakness i can find is the lack of proper baselines. The authors test against several, but none are current state-of-the-art. Their choice of models are primarily GANs, but their contribution is a diffusion model. They should compare against other diffusion models.

They include a ControlNet Encoder, but don't compare against the original setup with stable diffusion. This research would be far more relevant if it were compared to the state of the art from which they clearly draw inspiration.

---

### Official Review · Reviewer_ZWgE · 2025-11-03

**Soundness:** 3
**Correctness:** 3
**Rating:** 5
**Confidence:** 4

**Summary:**

This extended abstract considers the problem of generating realistically looking face images from sketches (think about a police sketch of a fugitive).

The suggested approach takes a sketch and generates a first coarse image using a GAN. This coarse image along with some variations such as a noisy version, various masked versions, and a segmentation are then used together with the original sketch in a conditional diffusion model that is guided by a ControlNet and Dino-v2 embeddings that are also derived from the initial coarse image.

The proposed method is then evaluated against multiple baseline models using various metrics on the CUFS dataset. Additional qualitative comparisons and an ablation study complement the evaluation.

**Strengths:**

- Conditional diffusion and sketch-to-face image synthesis are relevant topics.
- The paper is mostly well written.
- Overall, the proposed solution makes sense.
- The initial experimental results are promising.

**Weaknesses:**

- It is not clear how and why the Dino-v2 embeddings are helpful.
- It is not clear if a noisy version of $I_\text{coarse}$ or $I_\text{out}$ is used as $I_t$.
- Since this approach can be used to generate realistic images of fugitives, I am missing sections about limitations and broader impacts. What if the model fails and provides a memorized face? What are the ethical implications?
- Figure 1 lacks detail. See my questions below.


Some questions:
- Why is a noisy version $I_t$ of $I_\text{coarse}$ needed?
- Line 066 states that for $I_t$, the input imaged is used with added noise. What is the input image? To me, this suggests $I_\text{coarse}$, but in Figure 1 it looks like a noisy version of $I_\text{out}$ was used. If this is done for training, how would a production pipeline then look like?
- Is the segmentation computed on $I_\text{out}$ or on $I_\text{coarse}$?
- Where does the segmentation $I_\text{seg}$ come from?
- Why are experimental details like the optimizer choice and learning rate provided but actually important things like the segmentation model and the architecture of the Diffusion Denoising Unet not?

Minor comments:
- What is PBE (line 037)? $\implies$ a PBE-based baseline
- inconsistent use of dashes, e.g., lines 032, 033, etc.
- DINO-V2 vs Dino-v2
- The style of all $I_\text{something}$ in Figure 1 is different.
- Figure 1 could be arranged in a better way. I.e., by moving the $I_\text{coarse}$ to Dino-v2 part in Stage 2 down, one can draw a line directly from $I_\text{coarse}$ and also to $I_t$. Likewise, if $I_s$ is moved above, it can come from $I_2$ from Stage 1.
- controlNet vs ControlNet
- Lines 109-110: Figure 3 is only qualitative, not quantitative.
- Line 113: No such numbers are provided.

---

### Decision · Program_Chairs · 2025-11-05

**Decision:**

Accept

**Comment:**

The abstract is of interest to the community and should be presented at the conference.